# Measuring Health Literacy in Southern Italy: A cross-sectional study

**Sara Schiavone**[ID]**, Francesco Attena**[ID]*

Department of Experimental Medicine, University of Campania "Luigi Vanvitelli", Naples, Italy

* francesco.attena@unicampania.it

## Abstract

### Introduction

Health Literacy (HL) is an important determinant of individual health. Limited HL is an increasing problem affecting the general population. This study aims to assess the level of HL in patients attending outpatient medical facilities in general medicine located in Naples and Caserta and investigate the association of HL with health behaviours and health status.

### Materials and methods

The study involved patients attending outpatient medical facilities in general medicine. The questionnaire had four sections–the sociodemographic information, the 16-items version of the European Health Literacy Survey questionnaire, the general self-efficacy scale (GSE) and the health status scale (EQ-VAS). Univariate and multivariate analyses were performed to investigate the sociodemographic determinants of HL. The Pearson correlation coefficients were determined to compare HL with health behaviours (GSE) and health status (EQ-VAS).

### Results

The study showed that 61.6% of 503 patients had a low level of HL. After the multivariate analysis, HL was found to be higher among patients with higher education level and general self-efficacy score ≥30. There were no differences in HL between the age groups and people with or without chronic diseases. HL was stronger correlated with GSE than with EQ-VAS (0.53 vs 0.27).

### Conclusion

This is the first study on HL for Southern Italy. It showed a low level of HL. As the sample was not representative of the reference population, we cannot derive a corresponding conclusion for the general population of Southern Italy. Therefore, more data in Italy are needed to plan actions for improving HL.

**Data Availability Statement:** All relevant data are within the manuscript and its Supporting Information files.

**Funding:** The data presented in this work was partly funded by an Erasmus grant from a collaborative project involving both FA and SS. The

funder had no role in study design, data collection and analysis, decision to publish, or preparation of the manuscript. Please confirm this update is appropriate and please remove the mention (partly funded by an Erasmus grant) from your Acknowledgements.

**Competing interests:** The authors have declared that no competing interests exist.

## Introduction

Health Literacy (HL) is an important determinant of individual health [1–3]. HL refers to 'the knowledge, motivation and competence to access, understand, appraise and apply health information in order to make judgement and take decisions in everyday life concerning health care, disease prevention and health promotion, to maintain or improve quality of life throughout the course of life' [1, 2, 4]. HL can be briefly defined as the capability to make informed health decisions in daily life [5]. A high level of HL means being able to make reasonable judgement and explain the health problem and personal concerns that make a medical consultation necessary [6]. In addition, HL provides the capability to change one's beliefs if necessary [6]. Conversely, limited HL is an increasing problem affecting the general population [7–15]. A low level of HL leads to the misuse of available health resources and inappropriate access to care, which is defined as the degree of adaptation between patient skills and health-care system requirements [12, 16, 17].

HL covers three different areas–health care, disease prevention and health promotion. According to the European Health Literacy Survey, almost every second, EU citizen has limited HL and therefore perceived difficulties in accessing, understanding and using health information [4, 8, 18, 19].

HL has gained increasing attention in public health research, as well as health services reform processes, as an essential determinant of individual health and health service use [15, 19–24]. The evaluation of HL in Italy is important to ensure the sustainability of the health-care service system [12, 21, 25].

General practitioners, also called family doctors, are the first contact for health concerns in the Italian health system. They have different responsibilities for their patients: solving health problems, controlling adherence to treatment, ensuring continuity of care, identifying the correct path within the complexity of the health service and carrying out health education for promoting health and well-being. Therefore, the general practitioner should be the first contact person in the health literacy process.

In Italy, as in other European countries, the population of old people and therefore the prevalence of chronic diseases is rapidly increasing [26]. This indicates the need for a more complex health service, as several studies have suggested that HL decreases with age [27–29]. The decline in HL in older age groups is associated with decreasing cognitive function and potential health impairments [8, 17]. Globally, HL has been an important topic in public health research over the past decades [30]. However, in Italy, data on HL among the general population are still scarce [11, 12, 31].

This study aims to assess the level of HL in patients attending outpatient medical facilities in general medicine located in Naples and Caserta using the 16-items version of the European Health Literacy Survey questionnaire (HLS-EU-Q16) and investigate the association of HL with health behaviours and health status. The HLS-EU-Q16 questionnaire was recently developed and validated in the Italian language with a Cronbach's alpha of 0.799 [31]. This short version made measuring HL in the general population easier [11, 31].

## Materials and methods

### Study design and setting

This study used a cross-sectional design in which data collection was carried out between April 2019 and May 2019. Fifteen general practitioners working in outpatient medical facilities in general medicine located in Naples and Caserta were randomly selected from the register of the Local Health Authority of Naples and Caserta. Five of the selected general practitioners did not consent to participate. The participating patients were citizens enrolled in these outpatient

medical facilities. The study was, therefore, conducted in the waiting room of the remaining 10 medical facilities (5 in Naples and 5 in Caserta). The outpatient medical facilities in general medicine normally open five times a week, between mornings and afternoons, at the discretion of the general practitioner based on the total number of patients, up to a maximum of 1,500 patients for each general practitioner. According to the regulations of the National Healthcare Service and the Constitution of the Italian Republic, every Italian and foreign resident over the age of 18 years has to be registered in an outpatient medical facility in general medicine [5]. Ethical approval was obtained from the Ethics Committee of the University of Campania 'Luigi Vanvitelli' (Prot. N 302/2019).

## Sample size

The sample size was estimated to be at least 400 subjects, assuming a 50% of expected prevalence of the main outcome (high/low level of HL), with precision of 5% and level of significance of 95%.

## Study participants and data collection

The study population comprised patients attending outpatient medical facilities in general medicine. The inclusion criteria were patients who could speak Italian and were at least 18 years old. The patients waiting for a medical consultation were advised of the research project by the secretary of the general practitioner. They were informed that participation was voluntary and that they could withdraw from the study at any time with no subsequent consequences. The medical researcher was available to the participants to answer questions relating to the protocol. The questionnaire was self-administered and the participants were asked to sign the informed consent form. The exclusion criteria were patients with cognitive impairment, severe psychiatric diseases and end-stage diseases. All the data were collected anonymously.

## Questionnaire

The questionnaire had sections such as sociodemographic information (age, sex, education level and chronic diseases), the HLS-EU-Q16, the general self-efficacy scale (GSE), the Euro-Qol visual analogue scale (EQ-VAS) [31–33] and an Italian version of the PEN-13 (not evaluated because still under the Italian validation) [34]. The total number of questions was 44. The average duration to fill the questionnaire was 20 minutes.

The HLS-EU-Q16 contained 16 items, which was used to measure HL in the study populations. Each of the respondents was asked to give their opinion on a 4-point Likert scale–'very difficult', 'difficult', 'easy' and 'very easy'. The questionnaire covered the conceptual model proposed by Sorensen et al. [4] by investigating the ability of individuals to access or obtain, understand, process and use health information.

Beside HLS-EU-Q16, two other questionnaires have been used to investigate the association of HL with health behaviours and health status.

The GSE is a standardised measurement tool consisting of 10 items that capture the overall self-efficacy in a one-dimensional way. Each of the respondents was asked to give their opinion on a 4-point Likert scale–'Not at all true', 'Hardly true', 'Moderately true', 'Exactly true' [32].

EQ-VAS is a visual analogue scale for assessing a subject's view on global health status from 0 to 100 [33].

## Statistical analysis

Descriptive analysis was performed to evaluate the level of HL in the study population. To calculate the score of the HLS-EU-Q16, the answers were dichotomised into two categories with

two scores–easy (easy or very easy) and difficult (difficult or very difficult). The HL score is a sum score and three levels have been defined–inadequate HL (0–8), problematic HL (9–12) and adequate HL (13–16). The categories were dichotomised into adequate and not adequate (inadequate and problematic). This approach has been utilised previously [7, 11, 20, 31]. The category with adequate HL is assumed to have high level of HL, the category with not adequate is assumed to have low level of HL.

Age was recorded in years and categorised into three groups (18–45 years, 46–65 years and ≥65 years). Education level was assessed using the International Standard Classification of Education (ISCED 2011), which allows for cross-national comparisons of education levels and is dichotomised into two groups (ISCED 0–2 and ISCED 3–8) [35].

To investigate the sociodemographic determinants of HL, univariate analysis was performed and the results were expressed in terms of odds ratio, confidence interval and *p*-value. *P*-value ≤ 0.05 was considered statistically significant. Variables that showed a *p*-value ≤ 0.25 in the univariate analysis were included in the multivariate logistic regression.

The Pearson correlation coefficients for continuous variables were determined to compare HL level with health behaviours (from GSE) and health status (from EQ-VAS). Correlations with a coefficient from 0.1 to below 0.3 were considered as low, from 0.3 to below 0.5 as medium and from 0.5 and above as strong [36]. We assumed a high positive correlation between the HL level and the GSE score. This assumption is based on the fact that believing in the achievement of desired health outcomes leads a more active and self-determined life [12, 37–39]. We assumed a moderate positive correlation between HL score and the assessment of personal health status using EQ-VAS. Poor HL skills were associated with lower self-perceived health status [12]. Patients with low HL reported higher hospitalisation rates and greater use of health services [12, 23, 40]. In contrast, people with high HL skills are less likely to smoke or consume alcohol, and, generally, have a better health status [12].

The statistical analysis was carried out using the IBM Statistical Package for Social Science (version 21).

## Results

### Sociodemographic characteristics

In total, 503 patients completed the questionnaire and 42 (7.7%) declined to participate. The sociodemographic characteristics of the participants are reported in Table 1. There were more females (60.2%) than males (39.0%); most of the patients were more than 45 years (67.8%) and 62.8% had a high level of education. Among the patients, 50.7% had one or more chronic diseases.

### Health Literacy

The analysis of the 503 questionnaires and the dichotomisation of the responses into two categories showed that 61.6% of the patients had a low level of HL (Table 2). In Table 2, the sociodemographic characteristics were reported in comparison to the level of HL. There was no difference in HL between the males and the females. We found a higher level of HL among younger patients and those with a higher level of education. Moreover, patients who had no chronic diseases showed a higher level of HL (48.0%) compared to patients with one or more chronic diseases (30.6%). In the multivariate analysis, two variables remained associated with high HL: high education level and general self-efficacy score ≥30. Using a stratified analysis, we found that level of education was the main confounder of the association between age and HL because a lower level of education was more frequent in older patients (Table 3).

**Table 1. Socio-demographic characteristics of the participants.**

| Socio-demographic characteristics | n | % |
|---|---|---|
| **Gender** | | |
| Male | 196 | 39.0 |
| Female | 303 | 60.2 |
| Missing | 4 | 0.8 |
| **Age** | | |
| Mean (Standard deviation) | 52,5 (16,7) | |
| Range | 18–88 | |
| 18–45 | 150 | 29.8 |
| 46–65 | 208 | 41.4 |
| ≥ 65 | 133 | 26.4 |
| Missing | 12 | 2.4 |
| **Education Level** | | |
| ISCED§ 0–2 | 184 | 36.6 |
| ISCED 3–8 | 316 | 62.8 |
| Missing | 3 | 0.6 |
| **Chronic Diseases** | | |
| Yes | 255 | 50.7 |
| No | 229 | 45.5 |
| Don't know | 16 | 3.2 |
| Missing | 3 | 0.6 |
| Total | 503 | 100 |

§International Standard Classification of Education.

## Pearson correlation

We correlated the HLS-EU-Q16 score with GSE and EQ-VAS scores. The coefficient of correlation between the scores is shown in Table 4. HL was stronger correlated to general self-efficacy than the assessment of personal health status. In particular, a strong positive correlation between the HL score and the GSE score was found (0.53), while a weak positive correlation between the HL score and the assessment of personal health status using EQ-VAS was found (0.27). A weak positive correlation was also found between the assessment of personal health status and general self-efficacy (0.28).

## Discussion

The study investigated HL level among 503 patients attending medical facilities in general medicine located in Naples and Caserta, in Southern Italy. We achieved a high response rate (92.3%) and a high level of completion of the questionnaire because the medical researcher was available to provide information to patients. Although not representative of the reference population, our analysis revealed that 38.4% of the participants had a high level of HL.

Findings from two other studies conducted in Italy using a representative sample, reported a higher level of HL using the HLS-EQ-Q47 in one [12], whereas the other study [11] showed a lower level using the HLS-EQ-Q16.

The scarcity of national studies on this topic did not allow us to make a valid comparison between North and South Italy. We considered comparing these two areas important because the imbalances between the north and the south are very high. Northern Italy is more industrialised, healthier and wealthier than Southern Italy and the National Health Service in Northern

**Table 2. Sociodemographic characteristics and health behaviours disaggregated for Health Literacy.**

| | | Low Health Literacy | High Health Literacy | crude *p*-value | Adjusted *p*-value* | Adjusted Odds Ratio* |
|---|---|---|---|---|---|---|
| **Age** | | | | | | |
| | ≥ 65 | 102 (76.7%) | 31 (23.3%) | | | 1 |
| | 46–65 | 126 (60.6%) | 82 (39.4%) | <0.001 | 0.508 | 1.3 (C.I. 0.7–2.4) |
| | 18–45 | 77 (51.3%) | 73 (48.7%) | | | 1.5 (C.I. 0.7–2.9) |
| | Total | 305 (62.1%) | 186 (37.9%) | | | |
| **Sex** | | | | | | |
| | Female | 188 (62.0%) | 115 (38.0%) | | | |
| | Male | 120 (61.2%) | 76 (38.8%) | 0.854 | - | - |
| | Total | 308 (61.7%) | 191 (38.3%) | | | |
| **Education Level** | | | | | | |
| | ISCED§ 0–2 | 148 (80.4%) | 36 (19.6%) | | | 1 |
| | ISCED 3–8 | 160 (50.6%) | 156 (49.4%) | <0.001 | <0.001 | 1.2 (C.I. 1,1–1,3) |
| | Total | 308 (61.6%) | 192 (38.4%) | | | |
| **Chronic Diseases** | | | | | | |
| | Yes | 177 (69.4%) | 78 (30.6%) | | | 1 |
| | No | 119 (52.0%) | 110 (48.0%) | <0.001 | 0.714 | 0.9 (C.I. 0.7–1,2) |
| | Total | 296 (61.2%) | 188 (38.8%) | | | |
| **GSE score** | | | | | | |
| | ≤ 29 | 244 (74.6%) | 83 (25.4%) | | | 1 |
| | ≥ 30 | 64 (36.8%) | 110 (63.2%) | <0.001 | <0.001 | 3.8 (C.I. 2.5–5,8) |
| | Total | 308 (61.5%) | 193 (38.5%) | | | |
| **Total** | | 310 (61.6%) | 193 (38.4%) | | | |

* Multivariate logistic regression (in the model, the following variables with a *p*≤0.25 have been included: age, education level, chronic diseases and self-efficacy score).
§ International Standard Classification of Education.

**Table 3. Stratified analysis between Health Literacy and age by education level.**

| | | Age | | | | | | | |
|---|---|---|---|---|---|---|---|---|---|
| | | 18–45 | | >45 | | Total | | | |
| *Education Level* | | N | % | N | % | N | % | RR* | *p*-value |
| ISCED§ 0–2 | Low HL | 22 | 73.3 | 125 | 82.8 | 147 | 81.2 | 1.11 (C.I. 0.91–1.36) | 0.30 |
| | High HL | 8 | 26.7 | 26 | 17.2 | 34 | 18.8 | | |
| ISCED 3–8 | Low HL | 55 | 45.8 | 103 | 54.5 | 158 | 51.1 | 1.14 (C.I. 0.96–1.37) | 0.16 |
| | High HL | 65 | 54.2 | 86 | 45.5 | 151 | 48.9 | | |
| Total | Low HL | 77 | 51.3 | 228 | 67.1 | 305 | 62.2 | 1.23 (C.I. 1.08–1.41) | <0.00 |
| | High HL | 73 | 48.7 | 112 | 32.9 | 185 | 37.8 | | |

§ International Standard Classification of Education.
* Relative risk.

**Table 4. Pearson's correlation among HL score, GSE score and EQ-VAS scale.**

| | EQ-VAS Scale | GSE score | HL score | *p*-value |
|---|---|---|---|---|
| **EQ-VAS** | 1 | 0.28 | 0.27 | <0.001 |
| **GSE** | 0.28 | 1 | 0.53 | <0.001 |
| **HL score** | 0.27 | 0.53 | 1 | <0.001 |

Italy is faster, richer and has better quality [41–43]. Consequently, patients in Northern Italy may have a higher HL than those in the South. If this was true, interventions to improve HL should be differentiated between these two geographical areas.

In Europe, Sorensen et al. [4] analysed HL level in eight countries (Austria, Bulgaria, Germany, Greece, Ireland, Netherlands, Poland and Spain). The highest level of HL was found in the Netherlands and the lowest level of HL in Bulgaria (respectively 71.4% and 37.9% of sufficient/excellent HL).

Many studies have shown that low level of HL is associated with poorer health outcomes, older people, lower education level, poorer self-rated health, limited use of preventive health services, increased hospital visits and higher mortality rates as well as inferior physical and mental health [19, 23, 24, 44–48].

Three sociodemographic characteristics (education level, age and chronic diseases) and two scales (GSE and EQ-VAS) were included in our analysis. Multivariate analysis showed that only the education level was associated with HL among the three sociodemographic characteristics. This is consistent with other studies [2, 4, 5, 10, 20, 49–53] and the World Health Organisation's statement that consider people with low level of education as a vulnerable group with a scarce HL [2].

Health behaviour measured by GSE and health status measured by EQ-VAS were positively correlated with HL as expected [2, 53, 54].

Health Literacy and self-efficacy are important factors contributing to health promotion behaviour. Therefore, it would be desirable for each person to have both high HL and high GSE. In our sample, these two factors were correlated, but this correlation was stronger when both factors were consistent (48.5% both low vs 21.9% both high). Therefore, only 21.9% of the respondents in our sample seem to have had means for health promotion behaviour.

In our healthcare system, general practitioners might actively promote HL of their target group by providing information, advice, education and guidance. Their role should be important for the creation of an appropriate database to develop health literacy promotion strategies and to plan evidence-based interventions. However, the general practitioners included in our study would appear to have had little influence in promoting the HL of their patients.

Limitation. The main limitation of the study is related to its external and internal validity. Firstly, the respondents were a selected group of patients waiting in outpatient medical facilities in general medicine, and not a sample of the general population. Therefore, their HL level may be different from the general population. Secondly, they completed the questionnaire after the secretary of the general practitioner gave them the opportunity to participate in the study. They voluntarily took part in the study but the presence of the medical researcher during the completion of the questionnaire might have caused them not to answer the items truthfully, rather in a certain desired direction or under the psychological pressure of consulting with their general practitioner. It is also possible that patients might have completed the questionnaires hastily without paying particular attention because they were probably focused on seeing their general practitioner and did not want to miss their turn.

## Conclusion

This study reports the first results on HL for Southern Italy in outpatient medical facilities in general medicine. It showed a low level of HL. As the sample was not representative of the reference population, we cannot derive a corresponding conclusion for the general population of Southern Italy. Despite the important role of the general practitioners, they would appear to have had little influence in promoting the HL of their patients. Therefore, more data about HL in this area and in Italy are needed for a better understanding of the situation in the country and to plan actions to improve HL in our population.

## Supporting information

**S1 Data.**
(SAV)

## Acknowledgments

The data presented here originate from a collaborative project led by Monika A. Rieger, Achim Siegel (University Hospital Tübingen, Institute of Occupational and Social Medicine and Health Services Research), the corresponding author Francesco Attena and the first author Sara Schiavone. In this project, amongst others, an Italian version of the measurement tool 'Patient Enablement Scale– 13 Items' (PEN-13) was developed and validated. The authors gratefully acknowledge the general practitioners who agreed to participate. A special thanks to Anna Ehmann, Dr. Achim Siegel and Prof. Monika A. Rieger for their professional advice and for their methodology suggestions.

## Author Contributions

**Conceptualization:** Sara Schiavone.

**Data curation:** Sara Schiavone.

**Formal analysis:** Sara Schiavone.

**Investigation:** Sara Schiavone.

**Methodology:** Francesco Attena.

**Project administration:** Francesco Attena.

**Software:** Sara Schiavone, Francesco Attena.

**Supervision:** Francesco Attena.

**Validation:** Francesco Attena.

**Writing – original draft:** Sara Schiavone.

**Writing – review & editing:** Francesco Attena.

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
