## [Decision Letter · Decision Letter 0]

29 Apr 2020

PONE-D-20-09482

Measuring Health literacy in South Italy: A Cross-sectional Study

PLOS ONE

Dear Dr. Attena,

Thank you for submitting your manuscript to PLOS ONE. After careful consideration, we feel that it has merit but does not fully meet PLOS ONE’s publication criteria as it currently stands. Therefore, we invite you to submit a revised version of the manuscript that addresses the points raised during the review process.

Both Reviewers agree that the study is valuable, though not innovative in its contribution to the literature. In addition, both Reviewers point to shortcomings in language that not only have to do with the use of English, but also more specifically with methodological correctness. Finally, a lack of content and technical scientificity of the discussions and conclusions are highlighted. Therefore, I invite the Authors to proceed with a careful and in-depth revision of the whole manuscript, drawing on the rich annotations of the Reviewers.

We would appreciate receiving your revised manuscript by Jun 13 2020 11:59PM. To enhance the reproducibility of your results, we recommend that if applicable you deposit your laboratory protocols in protocols.io, where a protocol can be assigned its own identifier (DOI) such that it can be cited independently in the future. For instructions see: http://journals.plos.org/plosone/s/submission-guidelines#loc-laboratory-protocols

We look forward to receiving your revised manuscript.

Kind regards,

Stefano Federici, Ph.D.

Academic Editor

PLOS ONE

Journal Requirements:

Additional Editor Comments (if provided):

Both Reviewers agree that the study is valuable, though not innovative in its contribution to the literature. In addition, both Reviewers point to shortcomings in language that not only have to do with the use of English, but also more specifically with methodological correctness. Finally, a lack of content and technical scientificity of the discussions and conclusions are highlighted. Therefore, I invite the Authors to proceed with a careful and in-depth revision of the whole manuscript, drawing on the rich annotations of the Reviewers.

Reviewers' comments:

Reviewer's Responses to Questions

**Comments to the Author**

1. Is the manuscript technically sound, and do the data support the conclusions?

Reviewer #1: Partly

Reviewer #2: Yes

2. Has the statistical analysis been performed appropriately and rigorously? 

Reviewer #1: Yes

Reviewer #2: Yes

3. Have the authors made all data underlying the findings in their manuscript fully available?

Reviewer #1: Yes

Reviewer #2: Yes

4. Is the manuscript presented in an intelligible fashion and written in standard English?

Reviewer #1: Yes

Reviewer #2: Yes

5. Review Comments to the Author

Reviewer #1: This is an innovative paper from South of Italy which explores health literacy in patients attending outpatients facilities. It is a cross sectional study that explores correlations between health literacy and demographic and health and self-efficacy variables. Overall it is not a big advancement of the field.

- Abstract, pg 2 ll 20. Please avoid the use of the term “residents”; it is a bit misleading as your study population is composed by outpatients attending general practices.

- Abstract, pg 2, 36-39. Conclusion can be improved providing more implications derived from the results of the study.

- Introduction, pg 3 ll 66. A paragraph concerning the use and the potential implication of the measurements of health literacy in the primary health context is needed.

- Introduction, pg 4 ll 69. The study aim can result misleading, if so stated. This study is not assessing the level of HL in Naples and Caserta, but rather it is assessing the level of HL in patients attending outpatient clinics located in Naples and Caserta.

- Materials and Methods, pg5 ll 77-78. Please provide a more clear description of the type of outpatients medical facilities considered (i.e type of services carried out and type of patients visited). Where they general practices?

- Materials and Methods, pg5 ll 94-95. The final number of the subjects included in the study population should be moved to the Results section.

- Materials and Methods, pg 5 ll-93-103. Please provide more details concerning on how patients were enrolled and how the survey was carried out. Where the patients waiting for a medical visit when first contacted? Was their doctor involved in any part of the recruitment or survey process? The survey was administered by the researchers or was self-administered?

- Materials and Methods, pg 5 ll-101-103. The sentence “The presence the medical researcher during the completion of the questionnaire led to a high degree of response” is a consideration that have to be moved and further explained in the discussion section.

- Materials and Methods, pg 5 ll105-107. Please provide details concerning the total number of questions in the survey and the average duration of the survey

- Materials and Methods, pg 5 ll106. The HLS-EU-Q16 used in the study was an already validated version? if yes, please state which version

- Materials and Methods, pg 5 ll 105. Please provide more details on how chronic diseases were investigated in the survey

- Results pg 7 ll 150. Please provide data concerning the total number of subjects invited to participate in the study, the number of those who consent to participate and the number of refusal

- Results. pg 7 ll 173-174. Education was not the only variable significantly associated with HL, also General Self efficacy Score resulted significantly associated in your multivariate analysis.

- Results. pg 7 ll 173-174 and Table 2. Please provide the Odds ratio of the variables included in the multivariate model

- Results. pg 7 ll 178-179. The sentence “The HLS-EU-Q16 questionnaire has been validated in Italy with a Cronbach’s alpha of 0.799,” is not a result of the study and should be moved in the discussion and/or in the introduction sections.

- Results, pg 8 ll 175-176. Elaborate a table to report the stratified analysis mentioned here.

- Results p 8 ll177-178. The comparison between the different locations of the outpatients facilities is not necessary as the sample was not designed to be representative of the resident populations of the municipalities of Naples and Caserta, therefore results of this analysis provide little insights. Should the authors prefer to keep this comparison, data concerning these variable have to be reported in table 1 and 2, and the variable should be considered in the univariate and multivariate analyses.

- Discussion, pg 10 ll203-205; pg 11 ll 214-216. Direct comparison with data derived from population based sample should be avoided; the design of your study cannot allow any kind of comparison with other HL level. It is important that like is being compared with like. If different sampling strategies were used then the differences are likely to be sampling variations, not any population level differences. If the data are not comparable then the findings should not be compared.

- Discussion, general comment on the section. The authors need to consider internal and external validity. Given the sampling, and potential for misleading findings, the paper should mainly focus on results that arguably have internal validity – i.e., the antecedent analysis. Furthermore, I suggest to consider discussing the possible implications that the measurement of HL may have in the context of primary health care services

- Discussion, pg 11 217-224. The meaning of this paragraph in not clear, please rephrase it

- DIscussion, pg 11 ll 230-232. The mentioned “Medical doctor” is referred to the doctor with which they had the visit (e.g. the general practitioner) ? or the medical researcher? Please clarify the recruitment and survey process here and in the method section

- Discussion, please discuss the meaning and implications of the association between HL and the General Self-Efficacy Score

- Conclusion.pg 11-12 ll 240-242. Conclusion section have to be expanded to report the main findings of the study and their implications for primary care practices and research

- General comment: please carefully review the manuscripts for typo and grammar errors. For instance”

Reviewer #2: Generally, the article provides a good insight into the level of Health Literacy a sample in Italy. The authors did a good job working on an unexplored area in Italy. My comments are stratified by section down below.

The article is well reported and understandable, however some language modifications are recommended

Results:

Line 168: the concept of "low level of HL" is not introduced and identified until the results section

In table 2, HL levels were classified into "high" and "low", whereas the dichotomization were made into "adequate" and "not adequate" i.e problematic and inadequate earlier in the methodology section. The definitions were not clear as to what constitute low HL; does it mean inadequate only or inadequate and problematic combined together?

Similarly, the term “sufficient HL” was used in line 203 although this was referred to as "high HL" earlier in the table. Similarly, the term “sufficient HL” was used in line 203

I recommend to unify the terms used across the paper starting from the methodology and be specific with the definitions.

Line 191: "Indeed" is informal word to use.

Discussion:

Line 210-213: language needs to be improved

I’d like to read about your view of the implications of the study

Thank you for your time and effort

6. PLOS authors have the option to publish the peer review history of their article (what does this mean?). If published, this will include your full peer review and any attached files.

Reviewer #1: No

Reviewer #2: No

---

## [Author Response · Author response to Decision Letter 0]

27 Jun 2020

27 June 2020

Re: Manuscript PONE-D-20-09482

Measuring Health literacy in South Italy: A Cross-sectional Study

PLOS ONE

Dear Editor,

Please find attached a revised version of our manuscript titled “Measuring Health Literacy in Southern Italy: a cross-sectional study” which we would like to resubmit in PLOS ONE. The following pages contain our responses to the academic editor and reviewers’ comments. For clarity, we present the comments of the reviewers in italics and respond to each comment point-by-point. Revisions to the text are shown using yellow highlighting for additions.

We also added an Acknowledgments section. 

Journal Requirements:

We provided PLOS ONE's style requirements and all relevant data within the paper.

REPLY TO EDITOR

We revised the language and the methodological correctness and improved the content and technical scientificity of the discussions and conclusions.

REPLIES TO REVIEWERS

R1- Abstract, pg 2 ll 20. Please avoid the use of the term “residents”; it is a bit misleading as your study population is composed by outpatients attending general practices.

REPLY: resident will be replaced with patients attending outpatient medical facility in general medicine. 

REVISED VERSION, pg 2 ll 26: This study aims to assess the level of HL of the residents in patients attending outpatient medical facilities in general medicine located in of Naples and Caserta…

R1- Abstract, pg 2, 36-39. Conclusion can be improved providing more implications derived from the results of the study.

REPLY: we changed the conclusion.

REVISED VERSION pg 3, ll 48-52: Conclusion. This is the first study on HL for the Southern Italy. and cannot represent the entire geographical area. It showed a low level of HL. As the sample was not representative of the reference population, we cannot derive a corresponding conclusion for the general population of Southern Italy. Therefore, more data about HL in this area and in Italy are needed to better understand the situation in our country and to plan actions to for improveing HL our system.

R1- Introduction, pg 3 ll 66. A paragraph concerning the use and the potential implication of the measurements of health literacy in the primary health context is needed.

REPLY: Done.

REVISED VERSION pg 4 ll 77-82: General practitioners, also called family doctors, are the first contact for health concerns in the Italian health system. They have different responsibilities for their patients: solving health problems, controlling adherence to treatment, ensuring continuity of care, identifying the correct path within the complexity of the health service and carrying out health education for promoting health and well-being. Therefore, the general practitioner should be the first contact person in the health literacy process.

R1- Introduction, pg 4 ll 69. The study aim can result misleading, if so stated. This study is not assessing the level of HL in Naples and Caserta, but rather it is assessing the level of HL in patients attending outpatient clinics located in Naples and Caserta.

REPLY: Done.

REVISED VERSION pg 4 ll 91-94: This study aims to assess the level of HL in Naples and Caserta in patients attending outpatient medical facilities in general medicine located in Naples and Caserta using the short 16-items version of the European Health Literacy Survey questionnaire (HLS-EU-Q16) and investigate the association of HL with health behaviours and health status.

R1- Materials and Methods, pg5 ll 77-78. Please provide a more clear description of the type of outpatients medical facilities considered (i.e type of services carried out and type of patients visited). Where they general practices?

REPLY: Done.

REVISED VERSION pg 5 ll 102 -108: Fifteen medical doctors general practitioners working in outpatient medical facilities in general medicine located in Naples and Caserta were randomly selected from the register of the Local Health Authority of Naples and Caserta. Five of the selected medical doctors general practitioners did not consent to participate. The participating patients were citizens enrolled in these outpatient medical facilities. The study was, therefore, conducted in the waiting room of the remaining 10 medical facilities (5 in Naples and 5 in Caserta).

R1- Materials and Methods, pg5 ll 94-95. The final number of the subjects included in the study population should be moved to the Results section.

REPLY: The final number of the subjects has been moved to the results section.

REVISED VERSION pg 8 ll 188: In total, 503 patients completed the questionnaire and 42 (7.7%) declined to participate.

R1- Materials and Methods, pg 5 ll-93-103. Please provide more details concerning on how patients were enrolled and how the survey was carried out. Where the patients waiting for a medical visit when first contacted? Was their doctor involved in any part of the recruitment or survey process? The survey was administered by the researchers or was self-administered?

REPLY: Done.

REVISED VERSION pg 6 ll 121 -133: The study population comprised 503 attendees of the patients attending outpatient medical facilities in general medicine. The inclusion criteria were patients who could speak Italian and were at least 18 years old. The patients waiting for a medical consultation were advised of the research project by the secretary of the general practitioner. They were informed that participation was voluntary and that they could withdraw from the study at any time with no subsequent consequences. The medical researcher was available to the participants to answer questions relating to the protocol. The questionnaire was self-administered and In addition, the participants were asked to sign the informed consent form. The exclusion criteria were patients with cognitive impairment, severe psychiatric diseases and end-stage diseases. All the data were collected anonymously. The patients completed the questionnaire after the staff of the outpatient medical facilities gave them the opportunity to participate in the study. The presence of the medical researcher during the completion of the questionnaire led to a high degree of response.

R1- Materials and Methods, pg 5 ll-101-103. The sentence “The presence the medical researcher during the completion of the questionnaire led to a high degree of response” is a consideration that have to be moved and further explained in the discussion section.

REPLY: The consideration has been moved and explained in the discussion section.

REVISED VERSION pg 12 ll 246-249: The study investigated HL level among 503 outpatients attending medical facilities of in general medicine located in Naples and Caserta, in Southern Italy. We achieved a high response rate (92.3%) and a high level of completion of the questionnaire because the medical researcher was available to provide information to patients. 

R1- Materials and Methods, pg 5 ll105-107. Please provide details concerning the total number of questions in the survey and the average duration of the survey.

REPLY: Done.

REVISED VERSION pg 6 ll 135-139: The questionnaire had sections such as sociodemographic information (age, sex, education level and chronic diseases), the HLS-EU-Q16, the general self-efficacy scale (GSE) and the EuroQol visual analogue scale (EQ-VAS) [31-33] and an Italian version of the PEN-13 (not evaluated because still under the Italian validation)[34]. The total number of questions was 44. The average duration to fill the questionnaire was 20 minutes.

R1- Materials and Methods, pg 5 ll106. The HLS-EU-Q16 used in the study was an already validated version? if yes, please state which version

REPLY: Done already in the Introduction section.

REVISED VERSION pg 4 ll 94-97: The short version HLS-EU-Q16 questionnaire was recently developed and validated in the Italian language with a Cronbach’s alpha of 0.799 [31]. This short version made measuring HL in the general population easier [11, 31] 

R1 - Materials and Methods, pg 5 ll 105. Please provide more details on how chronic diseases were investigated in the survey.

REPLY: The question about chronic diseases reported in the questionnaire was:

Ha una o piú patologie croniche (es.: diabete, ipertensione, insufficienza cardiaca, osteoporosis)?

No 

Si

Non lo so

Se Si, quali ____________________________

However, the medical researcher was willing to give further clarification and to explain the significance of a current chronic disease, as a disease that lasts for at least 3 months.

R1 - Results pg 7 ll 150. Please provide data concerning the total number of subjects invited to participate in the study, the number of those who consent to participate and the number of refusals.

REPLY: Done.

REVISED VERSION pg 6 ll 121-122: The study population comprised 503 attendees of the patients attending outpatient medical facilities in general medicine. 

REVISED VERSION pg 8 ll 188: In total, 503 patients completed the questionnaire and 42 (7.7%) declined to participate.

R1 - Results. pg 7 ll 173-174. Education was not the only variable significantly associated with HL, also General Self efficacy Score resulted significantly associated in your multivariate analysis.

REPLY: Done.

REVISED VERSION pg 9 ll 213-215: In the multivariate analysis, only associations between high education and high HL were confirmed two variables remained associated with high HL: high education level and general self-efficacy score ≥30.

R1- Results. pg 7 ll 173-174 and Table 2. Please provide the Odds ratio of the variables included in the multivariate model

REPLY: Done.

 Low Health Literacy High Health Literacy crude p-value Adjusted p-value* Adjusted

Odds Ratio*

Age 

 ≥ 65 102 (76.7%) 31 (23.3%) 1

 46 – 65 126 (60.6%) 82 (39.4%) <0.001 0.508 1.3 (C.I. 0.7 -2.4)

 18 – 45 77 (51.3%) 73 (48.7%) 1.5 (C.I. 0.7 -2.9)

 Total 305 (62.1%) 186 (37.9%) 

Sex 

 Female 188 (62.0%) 115 (38.0%) 

 Male 120 (61.2%) 76 (38.8%) 0.854 - -

 Total 308 (61.7%) 191 (38.3%) 

Education Level 

 ISCED§ 0-2 148 (80.4%) 36 (19.6%) 1

 ISCED 3-8 160 (50.6%) 156 (49.4%) <0.001 <0.001 1.2 (C.I. 1,1-1,3)

 Total 308 (61.6%) 192 (38.4%) 

Chronic Diseases 

 Yes 177 (69.4%) 78 (30.6%) 1

 No 119 (52.0%) 110 (48.0%) <0.001 0.714 0.9 (C.I. 0.7- 1,2)

 Total 296 (61.2%) 188 (38.8%) 

GSE score 

 ≤ 29 244 (74.6%) 83 (25.4%) 1

 ≥ 30 64 (36.8%) 110 (63.2%) <0.001 <0.001 3.8 (C.I. 2.5-5,8)

 Total 308 (61.5%) 193 (38.5%) 

Total 310 (61.6%) 193 (38.4%) 

Table 2. Sociodemographic characteristics and health behaviours disaggregated for Health Literacy

* Multivariate logistic regression (in the model, the following variables with a p≤0.25 have been included: age, education level, chronic diseases and self-efficacy score).

§ International Standard Classification of Education.

R1- Results. pg 7 ll 178-179. The sentence “The HLS-EU-Q16 questionnaire has been validated in Italy with a Cronbach’s alpha of 0.799,” is not a result of the study and should be moved in the discussion and/or in the introduction sections.

REPLY: The sentence has been moved in the introduction section, pg 4 ll 94-96.

R1- Results, pg 8 ll 175-176. Elaborate a table to report the stratified analysis mentioned here.

REPLY: Done. 

Table 3. Stratified analysis between Health Literacy and age by education.

 Age 

 18-45 >45 Total 

Education Level N % N % N % RR* p-value

ISCED§ 0-2 Low HL 22 73.3 125 82.8 147 81.2 1.11 (C.I. 0.91 – 1.36) 0.30

 High HL 8 26.7 26 17.2 34 18.8 

ISCED 3-8 Low HL 55 45.8 103 54.5 158 51.1 1.14 (C.I. 0.96 – 1.37) 0.16

 High HL 65 54.2 86 45.5 151 48.9 

Total Low HL 77 51.3 228 67.1 305 62.2 1.23 (C.I. 1.08 – 1.41) <0.00

 High HL 73 48.7 112 32.9 185 37.8 

§ International Standard Classification of Education.

*Relative risk.

R1- Results p 8 ll177-178. The comparison between the different locations of the outpatient facilities is not necessary as the sample was not designed to be representative of the resident populations of the municipalities of Naples and Caserta, therefore results of this analysis provide little insights. Should the authors prefer to keep this comparison, data concerning these variables have to be reported in table 1 and 2, and the variable should be considered in the univariate and multivariate analyses.

REPLY: The sentence has been deleted.

REVISED VERSION pg 10 ll 218-219: There were no differences between Napoli and Caserta in HL level and in the correlation of HL with other variables.

R1 - Discussion, pg 10 ll203-205; pg 11 ll 214-216. Direct comparison with data derived from population based sample should be avoided; the design of your study cannot allow any kind of comparison with other HL level. It is important that like is being compared with like. If different sampling strategies were used then the differences are likely to be sampling variations, not any population level differences. If the data are not comparable then the findings should not be compared. 

REPLY TO 203-205: We removed the word “comparing”, and we specified the non-comparability between studies.

REVISED VERSION pg 12 ll 249-255: Although not representative of the reference population, our analysis revealed that 38.4% of the participants had a sufficient HL high level of HL. Comparing this percentage with the only two Italian studies with these methods, but Findings from two other studies conducted in Italy using a representative sample, one reported a higher level of HL using the HLS-EQ-Q47 in one [12], whereas the other study [11] showed a lower level using the HLS-EQ-Q16.

REPLY TO 214-216: we rewrote the sentence.

REVISED VERSION pg 12 ll 264-270: Our results, compared with other European studies, showed a low level of HL [4, 10, 20]. As regard the study of Sorensen et al. [4], it is a multicentric study that utilised the HLS-EQ-Q47 in eight European countries, in which only Bulgaria had a lower HL than our study.In Europe, Sorensen et al. [4] analysed HL level in eight countries (Austria, Bulgaria, Germany, Greece, Ireland, Netherlands, Poland and Spain). The highest level of HL was found in the Netherlands and the lowest level of HL in Bulgaria (respectively 71.4% and 37.9% of sufficient/excellent HL).

R1 - Discussion, general comment on the section. The authors need to consider internal and external validity. Given the sampling, and potential for misleading findings, the paper should mainly focus on results that arguably have internal validity – i.e., the antecedent analysis. 

REPLY: comment about internal and external validity are reported in Limitation section. However, we better clarify these issues in that section.

REVISED VERSION pg 14 ll 294-305: Limitation. The main limitation of the study is related to its external and internal validity. Firstly, the respondents were a selected group of patients waiting in outpatient medical facilities in general medicine, and not a sample of the general population. Therefore, their HL level may be different from the general population. Moreover Secondly, they completed the questionnaire after the staff of the outpatient medical facilities secretary of the general practitioner gave them the opportunity to participate in the study. They voluntarily took part in the study but the presence of the medical doctor researcher during the completion of the questionnaire might have caused them not to answer the items truthfully, rather in a certain desired direction or under the psychological pressure of consulting with their medical doctor general practitioner. It is also possible that the patients might have completed the questionnaires hastily, without paying particular attention to the questions, because they were probably focused on seeing their doctor general practitioner and did not want to miss their turn. 

R1 - Furthermore, I suggest to consider discussing the possible implications that the measurement of HL may have in the context of primary health care services. 

REPLY: Done.

REVISED VERSION pg 14 ll 289-293: In our healthcare system, general practitioners might actively promote HL of their target group by providing information, advice, education and guidance. Their role should be important for the creation of an appropriate database to develop health literacy promotion strategies and to plan evidence-based interventions. However, the general practitioners included in our study would appear to have had little influence in promoting the HL of their patients.

R1 - Discussion, pg 11 217-224. The meaning of this paragraph in not clear, please rephrase it.

REPLY: Done.

REVISED VERSION pg 12 ll 271 -275: Many studies have shown that limited low level of HL is associated with one or more of these healthy and sociodemographic characteristics poorer health outcomes, older adults people, lower education level, poorer self-rated health, limited use of preventive health services, increased hospital visits and higher mortality rates as well as inferior physical and mental health [19, 23, 24, 44–48]. 

R1- Discussion, pg 11 ll 230-232. The mentioned “Medical doctor” is referred to the doctor with which they had the visit (e.g. the general practitioner) ? or the medical researcher? Please clarify the recruitment and survey process here and in the method section.

REPLY: Done.

REVISED VERSION pg 14 ll 297-302: Moreover Secondly, they completed the questionnaire after the staff of the outpatient medical facilities secretary of the general practitioner gave them the opportunity to participate in the study. They voluntarily took part in the study but the presence of the medical doctor researcher during the completion of the questionnaire might have caused them not to answer the items truthfully, rather in a certain desired direction or under the psychological pressure of consulting with their medical doctor general practitioner.

R1 - Discussion, please discuss the meaning and implications of the association between HL and the General Self-Efficacy Score.

REPLY: Done.

REVISED VERSION pg 13 ll 284-288: Health Literacy and self-efficacy are important factors contributing to health promotion behaviour. Therefore, it would be desirable for each person to have both high HL and high GSE. In our sample, these two factors were correlated, but this correlation was stronger when both factors were consistent (48.5% both low vs 21.9% both high). Therefore, only 21.9% of the respondents in our sample seem to have had means for health promotion behaviour.

R1 - Conclusion.pg 11-12 ll 240-242. Conclusion section have to be expanded to report the main findings of the study and their implications for primary care practices and research.

REPLY: Done.

REVISED VERSION pg 14 ll 308-315: This study reports the first results on HL for the Southern Italy in outpatient medical facilities in general medicine and cannot represent the entire geographical area. It showed a low level of HL. As the sample was not representative of the reference population, we cannot derive a corresponding conclusion for the general population of Southern Italy. Despite the important role of the general practitioners, they would appear to have had little influence in promoting the HL of their patients. Therefore, more data about HL in this area and in Italy are needed to for a better understanding of the situation in the our country and to plan actions to improve HL in our system population. 

- General comment: please carefully review the manuscripts for typo and grammar errors. For instance”

REPLY: All the manuscript has been reviewed by a mother tongue revisor. The edits are underlined in the tracked version.

Reviewer #2: Generally, the article provides a good insight into the level of Health Literacy a sample in Italy. The authors did a good job working on an unexplored area in Italy. My comments are stratified by section down below.

The article is well reported and understandable, however some language modifications are recommended

R2 - Results: Line 168: the concept of "low level of HL" is not introduced and identified until the results section.

REPLY: We introduced the concept in the methods.

REVISED VERSION pg 7 ll 158-161: The categories were dichotomised into adequate and not adequate (inadequate and problematic). This approach has been utilised previously [7, 11, 20, 31]. The category with adequate HL is assumed to have high level of HL, the category with not adequate is assumed to have low level of HL.

R2 - In table 2, HL levels were classified into "high" and "low", whereas the dichotomization were made into "adequate" and "not adequate" i.e problematic and inadequate earlier in the methodology section. The definitions were not clear as to what constitute low HL; does it mean inadequate only or inadequate and problematic combined together?

REPLY: We added the definition in the methods section pg 7 ll 158-161.

R2 - Similarly, the term “sufficient HL” was used in line 203 although this was referred to as "high HL" earlier in the table. I recommend to unify the terms used across the paper starting from the methodology and be specific with the definitions.

REPLY: We unified the terms in the methods and discussion section.

REVISED VERSION pg 12 ll 249-251: Although not representative of the reference population, our analysis revealed that 38.4% of the participants have a sufficient HL high level of HL.

R2 - Line 191: "Indeed" is informal word to use.

REPLY: we deleted Indeed and used In particular.

REVISED VERSION pg 11 ll 238-240: Indeed In particular, a high strong positive correlation between the HL score and the GSE score was found (0.53), while a low weak positive correlation between the HL score and the assessment of personal health status using EQ-VAS was found (0.27).

R2 - Discussion: Line 210-213: language needs to be improved.

REPLY: Done.

REVISED VERSION pg 12 ll 260-263: Consequently, though yet to be demonstrated, northern patients in Northern Italy may have a higher HL than those in the South. If this were was true, any interventions to improve HL would should have to be differentiated between these two geographical areas.

R2 I’d like to read about your view of the implications of the study.

REPLY: we improve the implication of the study in the conclusion section.

REVISED VERSION pg 14 ll 308-315: This study reports the first results on HL for the Southern Italy in outpatient medical facilities in general medicine and cannot represent the entire geographical area. It showed a low level of HL. As the sample was not representative of the reference population, we cannot derive a corresponding conclusion for the general population of Southern Italy. Despite the important role of the general practitioners, they would appear to have had little influence in promoting the HL of their patients. Therefore, more data about HL in this area and in Italy are needed to for a better understanding of the situation in the our country and to plan actions to improve HL in our system population. 

I hope that the revisions to the manuscript and the accompanying responses are acceptable, and that the manuscript is suitable for publication.

I look forward to hearing from you.

Yours sincerely,

Prof. Francesco Attena

Department of Experimental Medicine

University of Campania “Luigi Vanvitelli”

Via Luciano Armanni, 5

80138 Naples (Italy)

tel +39 081 5666012

e-mail francesco.attena@unicampania.it

---

## [Decision Letter · Decision Letter 1]

17 Jul 2020

Measuring Health Literacy in Southern Italy: a cross-sectional study

PONE-D-20-09482R1

Dear Dr. Attena,

We’re pleased to inform you that your manuscript has been judged scientifically suitable for publication and will be formally accepted for publication once it meets all outstanding technical requirements.

Kind regards,

Stefano Federici, Ph.D.

Academic Editor

PLOS ONE

Additional Editor Comments (optional):

Reviewers' comments:

Reviewer's Responses to Questions

**Comments to the Author**

1. If the authors have adequately addressed your comments raised in a previous round of review and you feel that this manuscript is now acceptable for publication, you may indicate that here to bypass the “Comments to the Author” section, enter your conflict of interest statement in the “Confidential to Editor” section, and submit your "Accept" recommendation.

Reviewer #1: All comments have been addressed

2. Is the manuscript technically sound, and do the data support the conclusions?

Reviewer #1: Yes

3. Has the statistical analysis been performed appropriately and rigorously? 

Reviewer #1: Yes

4. Have the authors made all data underlying the findings in their manuscript fully available?

Reviewer #1: Yes

5. Is the manuscript presented in an intelligible fashion and written in standard English?

Reviewer #1: Yes

6. Review Comments to the Author

Reviewer #1: (No Response)

7. PLOS authors have the option to publish the peer review history of their article (what does this mean?). If published, this will include your full peer review and any attached files.

Reviewer #1: **Yes: **Vieri Lastrucci

---

## [Editor Report · Acceptance letter]

28 Jul 2020

PONE-D-20-09482R1 

Measuring Health Literacy in Southern Italy: a cross-sectional study 

Dear Dr. Attena:

I'm pleased to inform you that your manuscript has been deemed suitable for publication in PLOS ONE. Congratulations! Your manuscript is now with our production department. 

Kind regards, 

on behalf of

Prof. Stefano Federici 

Academic Editor

PLOS ONE